# The burden of chronic diseases, disease-stratified exploration and gender-differentiated healthcare utilisation among patients in Bangladesh

**Rashidul Alam Mahumud** [1,2,3] *, **Jeff Gow** [3,4], **Md Parvez Mosharaf** [1,3], **Satyajit Kundu** [5,6], **Md. Ashfikur Rahman** [7], **Natisha Dukhi** [8], **Md Shahajalal** [9], **Sabuj Kanti Mistry** [10,11,12,13], **Khorshed Alam** [3]

1 Health Research Group, Department of Statistics, University of Rajshahi, Rajshahi, Bangladesh,
2 NHMRC Clinical Trials Centre, Faculty of Medicine and Health, The University of Sydney, Camperdown, New South Wales, Australia, 3 School of Business, Faculty of Business, Education, Law and Arts, University of Southern Queensland, Toowoomba, Australia, 4 School of Accounting, Economics and Finance, College of Law and Management Studies, University of KwaZulu-Natal, Durban, South Africa, 5 Global Health Institute, North South University, Dhaka, Bangladesh, 6 Faculty of Nutrition and Food Science, Patuakhali Science and Technology University, Patuakhali, Bangladesh, 7 Development Studies Discipline, Social Science School, Khulna University, Khulna, Bangladesh, 8 Human Sciences Research Council, Cape Town, South Africa, 9 Department of Public Health, School of Health and Life Sciences, North South University, Dhaka, Bangladesh, 10 Centre for Primary Health Care and Equity, University of New South Wales, Sydney, Australia, 11 ARCED Foundation, Dhaka, Bangladesh, 12 Department of Public Health, Daffodil International University, Dhaka, Bangladesh, 13 Brain and Mond Centre, The University of Sydney, Camperdown, New South Wales, Australia

* rashidul.icddrb@gmail.com

## Abstract

### Background

Chronic diseases are considered one of the major causes of illness, disability, and death worldwide. Chronic illness leads to a huge health and economic burden, especially in low- and middle-income countries. This study examined disease-stratified healthcare utilisation (HCU) among Bangladesh patients with chronic diseases from a gender perspective.

### Methods

Data from the nationally representative Household Income and Expenditure Survey 2016–2017 consisting of 12,005 patients with diagnosed chronic diseases was used. Gender differentiated chronic disease stratified-analytical exploration was performed to identify the potential factors to higher or lower utilisation of healthcare services. Logistic regression with step-by-step adjustment for independent confounding factors was the method used.

### Results

The five most prevalent chronic diseases among patients were gastric/ulcer (Male/Female, M/F: 16.77%/16.40%), arthritis/rheumatism (M/F: 13.70%/ 13.86%), respiratory diseases/ asthma/bronchitis (M/F: 12.09% / 12.55%), chronic heart disease (M/F: 8.30% / 7.41%),

**Data Availability Statement:** Bangladesh Household Income and Expenditure Survey (HIES) is conducted by the Bangladesh Bureau of

Statistics (BBS) with technical and financial support from the World Bank. This research was carried out using the 2016-2017 Bangladesh Household Income and Expenditure Survey. However, the BBS imposed legal restrictions that prevent the sharing of data publicly. Data can be shared upon request to the corresponding author with the permission of the BBS (Director General, Bangladesh Bureau of Statistics, dg@bbs.gov.bd, +88-02-5500-7056, www.bbs.gov.bd).

**Funding:** The author(s) received no specific funding for this work.

**Competing interests:** The authors have declared that no competing interests exist.

and blood pressure (M/F: 8.20% / 8.87%). Eighty-six percent of patients with chronic diseases utilised health care services during the previous 30 days. Although most patients received outpatient healthcare services, a substantial difference in HCU among employed male (53%) and female (8%) patients were observed. Chronic heart disease patients were more likely to utilise health care than other disease types, which held true for both genders while the magnitude of HCU was significantly higher in males (OR = 2.22; 95% CI:1.51– 3.26) than their female counterparts (OR = 1.44; 1.02–2.04). A similar association was observed among patients with diabetes and respiratory diseases.

## Conclusion

A burden of chronic diseases was observed in Bangladesh. Patients with chronic heart disease utilised more healthcare services than patients experiencing other chronic diseases. The distribution of HCU varied by patient's gender as well as their employment status. Risk-pooling mechanisms and access to free or low-cost healthcare services among the most disadvantaged people in society might enhance reaching universal health coverage.

## Background

Chronic diseases are the leading causes of death globally [1]. Estimates suggest that chronic diseases cause approximately 41 million deaths annually, equivalent to 71% of all deaths globally [2]. Each year, more than 15 million people die from a chronic disease who are aged between 30 and 69 years; 85% of these "premature" deaths occur in low- and middle- income countries [2]. The major chronic diseases are ischaemic heart disease, stroke, chronic obstructive pulmonary disease (COPD) and diabetes [3]. Notably, 75% of deaths from chronic diseases are associated with modifiable risk factors (e.g., tobacco use, physical inactivity, the harmful use of alcohol and unhealthy diets). The burden of chronic diseases adversely impacts individuals, communities, and families, resulting in health systems being overwhelmed and incurring large socioeconomic costs [2]. Therefore, investing in chronic disease detection, screening, treatment, and palliative care are vital components of an effective response to achieve the 2030 Agenda for Sustainable Development Goal (SDG) 3.4: reducing premature deaths by one-third by 2030 [4].

Bangladesh is a developing country that is undergoing both epidemiologic and demographic transitions, where the disease burden is shifting from an infectious disease dominated illness profile to a highly characterised chronic diseases profile, coupled with socioeconomic inequality and occurring in predominantly rural populations [5, 6]. Chronic diseases account for approximately 61% of total burden of disease and 54% of annual mortality in Bangladesh, with diabetes, cardiovascular disease, chronic respiratory disease, cancer, and stroke most common illnessess [7, 8]. The global burden of disease country profile for Bangladesh highlights a trend of increased chronic disease mortality due to stroke, ischaemic heart disease, chronic kidney disease, chronic pulmonary disease and diabetes [9]. A recent study suggested that the prevalence of double and triple burden of chronic disease in Bangladesh is 21.4% and 6.1%, respectively [10]. A survey of the non-communicable disease (NCD) risk factors survey in 2010 indicated that approximately 99% of population had at least one NCD risk factors [11]. In their disease risk factor survey in 2010, the WHO indicated that the death rate from only cardiovascular disease will be increased 21 times in Bangladesh by 2025 where a study

suggested 53.7% rural elderly population had chronic comorbid conditions [12]. Significant challenges to managing this situation exist such as a highly unregulated private health sector and a weak public health system must be addressed in conjunction with the ongoing increase in chronic diseases. With more than 70% of the population residing in rural areas, access to formal healthcare services are minimal due to an urban treatment bias and out-of-pocket expenditure is high with minimal health insurance coverage [13, 14]. HCU is often compounded by high treatment costs, limited access to proper care, inadequate or lack of infrastructure, and socioeconomic gaps [6]. Evidence also indicates that households with individuals with one or more chronic diseases face significantly higher financial risks [15, 16].

Adopting a gendered approach to chronic diseases management is an imperative, as men and women function different biologically and therefore face different health risks, experience different health system responses, and their health-seeking behaviours differ, all impacting their health outcomes as well as HCU [17, 18]. The literature shows that due to lower social status, limited education access, and economic vulnerability, women are disproportionately affected than men in their HCU [19]. In one Indian study, older women reported decreased HCU compared to their male counterparts, resulting in worse self-rated health and higher disability prevalence among them [20]. A study in Canada conversely found that women utilise healthcare services much more than men and spend more on healthcare [21]. A recent study found that women are lagging behind than their male counterparts in utilising inpatient care for chronic diseases such as diabetes, hypertension, chronic lung disease, depression, stroke, and asthma [21]. This study also identified males were willing to travel greater distances to access better-equipped healthcare facilities than women who tend to seek inpatient care at facilities near their homes.

There are still gender disparities in decision-making, roles and rights at home, and self-esteem when it comes to empowering women, which limits their access to healthcare in developing countries [20–22]. This is also true for Bangladesh, where men are often viewed as the head of households, decision-makers and are usually in charge of household resources and who typically decide on the women's health needs and where and when they should utilise healthcare services [20, 22]. To achieve SDG 5: Health and gender equality, it is imperative to ensure women have access to appropriate health care utilisation [20, 22]. However, there is scant information on existing gender disparities in utilising healthcare services among patients with chronic diseases in Bangladesh.

As a result, this study aimed to examine the gender perspective of HCU among patients with chronic diseases in Bangladesh.

## Methods

### Study design and data source

This cross-sectional study used data from a nationally representative survey, the Household Income and Expenditure Survey (HIES) in Bangladesh. The HIES is commonly used worldwide, especially in developing countries, to assess poverty levels and people's living standards. The HIES survey collects information about each household income, expenditure, consumption, health and social safety and other socio-economic aspects [23]. HIES in Bangladesh is a periodic cross-sectional survey conducted every five years by the Bangladesh Bureau of Statistics (BBS). The present study used data from the most recent HIES conducted in 2016–2017. Bangladesh Bureau of Statistics (BBS) has already validated the study settings and tested the reliability of the data [23]. The details of the study settings, questionnaire, and quality control measures have been described in the HIES 2016–17 report summary [23]. The HIES 2016–17 survey was based on an established protocol [23]. The HIES is a cross-sectional survey

conducted by the BBS in Bangladesh every five (5) years since 1973; throughout the period of implementation, the HIES tools have been thoroughly reviewed to address the validity and reliability of the results. In line with the objective of HIES survey, the HIES 2016–17 survey collected information under nine modules: 1) household information, 2) education, 3) health: illnesses and injuries, 4) economic activities and wage employment, 5) non-agricultural enterprises, 6) housing, 7) agriculture, 8) other assets and income, and 9) consumption. However, the objective of the current article was to investigate disease-stratified healthcare utilisation (HCU) among Bangladesh patients with chronic diseases from a gender perspective. Therefore, we only use the indicators pertaining to chronic disease and health service utilisation along with socio-demographic characterises of the participants. The HIES datasets are widely accepted and validated to produce scientific evidence. It is also used for monitoring the progress of poverty reduction and the Sustainable Development Goals (SDGs) indicators in Bangladesh.

## Sampling

A stratified, two-stage random sample design was adopted for the data collection. In the first stage, primary sampling units (PSUs) throughout the country from 20 strata (8 rural, 8 urban, and 4 metropolitan areas) were randomly selected to achieve national representation. A total of 2304 PSUs were selected using systematic random sampling from the list of 2011 Housing and Population Census enumeration areas. A PSU is usually a geographically constructed area, or a part of an area, called an enumeration area (EA), containing a number of households, created from the most recent population census. In the second stage, 20 households within each PSU were randomly selected (BBS, 2016). Using this sampling technique, a total of 46,080 households were included in HIES 2016–2017. Among the selected households, a total of 186,076 individuals were interviewed. Data collection was performed between April 1, 2016 and 31 March 2017. The survey objectives, sampling technique, survey design, survey instruments, measuring system, and quality control have been described elsewhere [23].

Participants selection criteria:

The participants for the current analysis were selected based on the following inclusion criteria: (1) an individual who suffered from any chronic illness in the last 12 months, (2) individual who received any treatment due to chronic illness in the last 30 days at the interview time, and iii) individual who received any treatment due to chronic illness in the last 30 days. Based on these inclusion criteria, a total of 12,005 participants were selected for analytical exploration in this study (Fig 1).

## Study variables

**Outcome measure.** This study considered 'patient's utilisation of healthcare services due to chronic illness as an outcome variable. As a measure of their HCU related to chronic illness, participants responded to questions that asked them about any type of medical treatment: "Have you sought any type of medical treatment related to your illness in the last 30 days?" Response options were recoded as dichotomous ('yes' if the patient received any type of medical treatment due to illness, or 'no' otherwise).

**Chronic diseases.** All health-related information was self-reported in the HIES survey. For conducting the analysis, we used chronic disease-related questions that were included in Module-3: Health (Illnesses and Injuries) of the primary survey [23]. For example, while collecting chronic disease-related information, the enumerators were instructed to ask the respondents, "Have you suffered from any chronic illness/disability in the last 12 months or more?" (if yes); then, participants were asked a second question "What chronic illness/

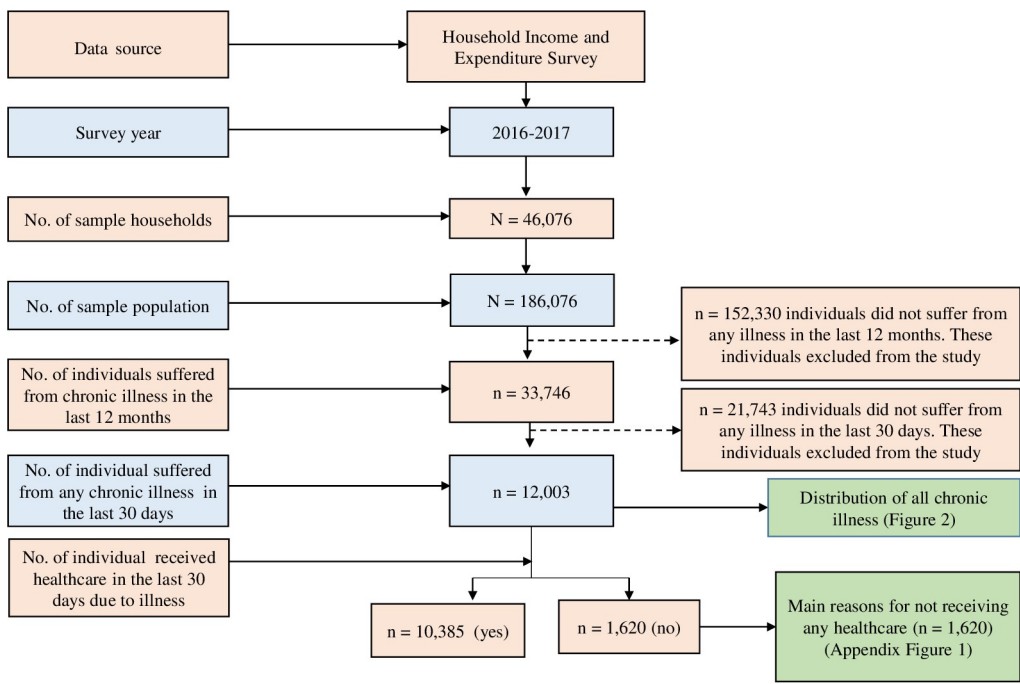

Figure 1. Sample selection

**Fig 1. Sample selection.**

disability are you suffering from?" with response options: 1) chronic fever, 2) Injuries/disability, 3) Chronic heart disease, 4) Respiratory Diseases/ Asthma/Bronchitis, 5) Diarrhoea/dysentery, 6) Gastric/ulcer, 7) Blood pressure, 8) Arthritis/Rheumatism, 9) Skin problem, 10) Diabetes, 11) Cancer, 12) Kidney diseases, 13) Liver Diseases, 14) Mental Health, 15) Paralysis, 16) Ear/ENT problem, 17) Eye problem, or 18 other (specify). Participants responded based on their disease diagnosis, experiences, symptoms of illness and course of treatment. For analysis, we recorded type of diseases based on the most reported but kidney diseases had a low frequency ones (Fig 2).

The explanatory variables considered in the study were demographic characteristics (gender, age, marital status, education, employment); type of chronic disease (e.g., chronic heart disease, respiratory diseases, gastric or ulcer, blood pressure, arthritis or rheumatism, diabetes, chronic fever, and other diseases); number of chronic comorbid conditions (one chronic condition, two chronic comorbid conditions and three or more chronic comorbid conditions); type of healthcare provider (public hospital, private clinic or hospital, pharmacy/dispensary, doctor's chamber, or others), type of healthcare services (inpatient or outpatient care); and location of the consulted healthcare provider (urban or rural).

## Statistical analysis

In the descriptive analyses *analysis*, characteristics of the study participants were expressed using frequencies, n (%). The dependent variable (i.e., utilisation of health care services: chronic illness patients who received any type of medical treatment due to chronic illness in the last 30 days) was characterised as a dichotomous measure. For the analytical exploration, the choice of estimation approach was informed by the nature of the outcome variables under consideration in each model. The logistic regression model was used to identify the potential

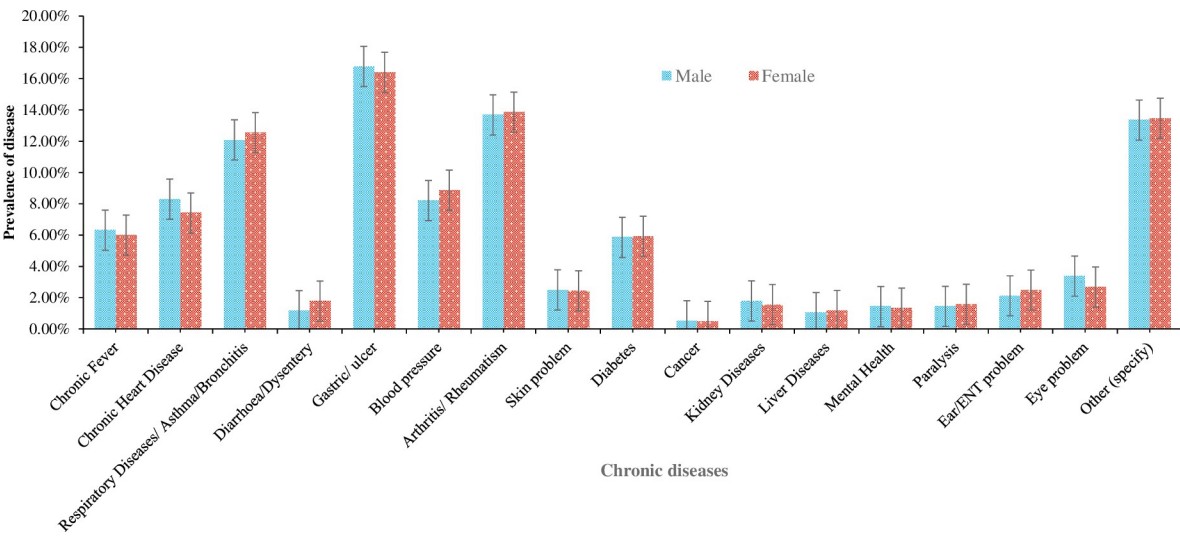

Figure 2. Distribution of chronic illness

**Fig 2. Distribution of chronic illness.**

factors that had a significant role in utilising healthcare services. The potential factors of utilising healthcare services owing to chronic illness were investigated using binary logistic regression models, with the results provided as odd ratios (OR) [i.e., exponential form of regression coefficient, OR = exp (beta)] and 95% confidence intervals. The regression model can be expressed as-

$$logit\ (Y_i) = \alpha + \beta_1 X_{1i} + \beta_2 X_{2i} + \ldots\ldots + \in_i$$

Where '$Y_i$' is the dichotomous outcome variable (i.e., utilisation of healthcare services at the last 30 days due to chronic illness), $\beta_1$, $\beta_2$, . . .. are the regression coefficients for the corresponding explanatory variables; $X_{1i}$, $X_{2i}$. . ...denote explanatory variables; and $\epsilon i$ is the error term.

$$\text{Utilisation of healthcare services } (Y_i) = \begin{cases} 0, & \text{if an individual did not receive healthcare services} \\ 1, & \text{if an individual received healthcare services} \end{cases}$$

To build the regression model, explanatory variables were selected based on published literature [16, 24–28], available variables of this dataset and explored bivariate relationships (unadjusted analysis) between variables. In our analysis, we investigated individual-level data to estimate disease-specific healthcare utilisation due to chronic illness by gender prospective.

The majority of the predictor variables were categorical in nature with two or more labels in this study. Therefore, an un-adjusted analysis was performed to find the association between outcome and the label of explanatory variables (Model 1). Our analyses were stratified by chronic diseases and gender prospective. For all diseases, the unadjusted explorations were expressed in Model 1, where most of the patients with chronic diseases were found to be significantly associated with lower or higher health care utilisation for both genders (all p ≤ 0.05). After step-by-step adjustment of independent confounding factors, we adjusted different variables into seven different models (i.e., variable related to type of health care was adjusted in Model 2). These were performed to present the variables that were significant in Models 2 to

Model 7, in order to understand how those variables are modified in the final model where all the variables were adjusted at the same time in the final Model 8.

For the independent variables, the category found to be least at risk of having patients' health care services related to chronic illness in the analysis was considered the reference category for constructing OR. We have followed standard 'svy' prefix command to address sampling weights. In addition, before running the final model, we checked for multicollinearity using Variance inflation factor (VIF) among the selected variables, no serious issues multicollinearity were found among the variables (all variables with VIF <5.00). The model was tested for sensitivity using the bootstrapping approach by resampling observations with 10,000 replications. Statistical significance was considered at $\leq$ 5% risk level. All data analyses were undertaken using the statistical software Stata/SE 14 (StataCorp, College Station, TX, USA).

### Ethical approval

The datasets were collected and made publicly available by the Bangladesh Bureau of Statistics (BBS). Since the de-identified data for this study came from secondary sources, this study did not require ethical approval.

## Results

### Participants' characteristics and distribution of chronic diseases

The total sample consisted of 12,005 patients ($\sim$ 50% of male) with one or more medically diagnosed chronic diseases (Table 1). 42% of male patients were young adults (18 to 45 years), and approximately 50% were married. However, one-third of patients had no formal education. Fifty-three percent of male patients were employed in the labour force, whereas only eight percent of female patients were employed. Most patients (94% of male and 92% of female) respondents reported at least one disability. The five most prevalent chronic diseases among multimorbid patients were gastric/ulcer (Male/Female (M/F):16.77% / 16.40%), arthritis/rheumatism (M/F:13.70% / 13.86%), respiratory diseases/asthma/bronchitis (M/F: 12.09% / 12.55%), chronic heart diseases (M/F: 8.30% / 7.41%), and blood pressure (M/F: 8.20% / 8.87%). We did not report the prevalence of all diseases due to low frequency (Fig 2). Most of the patients utilised outpatient health care services, with two-third of patients receiving health care from private hospitals or clinics usually in rural locations. 83% of patients reported 30 minutes (overall) or less waiting time to receive health care services.

### Distribution of health care utilisation (HCU)

The distribution of HCU due to chronic diseases by gender is presented in Table 1. Approximately overall 86% of patients utilised health care services in the last 30 days before the survey. 14% patients did not receive any healthcare services (Fig 3). The utilisation of health care reduced as patients aged. For instance, approximately one-third of the patients aged 18–35 years sought health care, which fell to around 5% among patients aged 65 years or more. Approximately 50% of married patients received any type of health care services (49.55% for males and 52.58% for females), which was among single participants. A significant difference in the prevalence of HCU was found between employed males (53.2; 95% CI: 51.80, 54.50) and female patients (7.81%, 95% CI: 7.11, 8.58). However, approximately 92% of unemployed female patients utilised health care compared to their unemployed male counterparts (47%). Patients who experienced gastric/ulcer sought health care most among both male (16.04%, 95% CI: 15.07, 17.06) and female respondents (15.78% [95% CI: 14.82, 16.80]), while chronic fever was found to be the lowest health care seeking disease for both genders.

**Table 1. Distribution of patient's characteristics and utilisation of health care services, by gender.**

| Participant characteristics | Male patients | | Female patients | |
|---|---|---|---|---|
| | Number of patients, n (%) | Utilisation of healthcare, % (95% CI) | Number of patients, n (%) | Utilisation of healthcare, % (95% CI) |
| **Age in years** | | | | |
| <18 years | 2,292 (38.28) | 38.21 (36.9, 39.54) | 2,210 (36.72) | 36.82 (35.51, 38.14) |
| 18–35 years | 1,748 (29.20) | 29.07 (27.85, 30.32) | 2,011 (33.42) | 33.28 (32.01, 34.57) |
| 36–45 years | 778 (12.99) | 12.86 (11.97, 13.80) | 715 (11.88) | 11.8 (10.95, 12.70) |
| 46–64 years | 859 (14.35) | 14.75 (13.81, 15.74) | 806 (13.39) | 13.24 (12.35, 14.19) |
| 65 or more | 310 (5.18) | 5.11 (4.54, 5.74) | 276 (4.59) | 4.87 (4.32, 5.49) |
| **Educational background** | | | | |
| No education | 2,208 (36.88) | 36.78 (35.48, 38.11) | 2,190 (36.39) | 36.24 (34.94, 37.56) |
| Up to primary | 1,739 (29.05) | 28.94 (27.72, 30.19) | 1,776 (29.51) | 29.58 (28.35, 30.84) |
| Secondary education | 1,608 (26.86) | 26.97 (25.78, 28.20) | 1601 (26.60) | 26.56 (25.38, 27.78) |
| Higher | 432 (7.22) | 7.31 (6.63, 8.05) | 451 (7.49) | 7.62 (6.93, 8.38) |
| **Marital status** | | | | |
| Currently married | 2,961 (49.46) | 49.55 (48.19, 50.91) | 3,164 (52.58) | 52.54 (51.18, 53.90) |
| Never married | 2,258 (37.72) | 37.69 (36.38, 39.02) | 1,657 (27.53) | 27.21 (26.02, 28.44) |
| Widowed/divorced/separated | 768 (12.83) | 12.76 (11.88, 13.70) | 1197 (19.89) | 20.25 (19.18, 21.36) |
| **Religion status** | | | | |
| Islam | 5,188 (86.65) | 86.81 (85.86, 87.71) | 5,222 (86.77) | 86.97 (86.03, 87.86) |
| Hinduism | 601 (10.04) | 9.95 (9.16, 10.79) | 595 (9.89) | 9.95 (9.16, 10.79) |
| Others | 198 (3.31) | 3.24 (2.79, 3.76) | 201 (3.34) | 3.08 (2.64, 3.59) |
| **Employed status** | | | | |
| Employed | 3,196 (53.38) | 53.2 (51.80, 54.50) | 491 (8.16) | 7.81 (7.11, 8.58) |
| Unemployed | 2,791 (46.62) | 46.8 (45.50, 48.20) | 5,527 (91.84) | 92.19 (91.42, 92.89) |
| **Any type of disability** | | | | |
| Yes | 5,636 (94.14) | 94.04 (93.36, 94.66) | 5,560 (92.39) | 92.42 (91.67, 93.11) |
| No | 351 (5.86) | 5.96 (5.34, 6.64) | 458 (7.61) | 7.58 (6.89, 8.33) |
| **Type of chronic illness** | | | | |
| Chronic heart disease | 497 (8.30) | 8.96 (8.22, 9.77) | 446 (7.41) | 7.78 (7.08, 8.54) |
| Respiratory diseases/ Asthma/ Bronchitis | 724 (12.09) | 12.53 (11.66, 13.46) | 755 (12.55) | 12.86 (11.97, 13.79) |
| Gastric/ulcer | 1,004 (16.77) | 16.04 (15.07, 17.06) | 987 (16.40) | 15.78 (14.82, 16.8) |
| Blood pressure | 491 (8.20) | 8.31 (7.59, 9.09) | 534 (8.87) | 8.83 (8.09, 9.64) |
| Arthritis/Rheumatism | 820 (13.70) | 13.69 (12.78, 14.65) | 834 (13.86) | 14.03 (13.11, 15.00) |
| Diabetes | 351 (5.86) | 6.23 (5.60, 6.92) | 357 (5.93) | 6.37 (5.74, 7.07) |
| Chronic fever | 378 (6.31) | 5.49 (4.91, 6.15) | 361 (6.00) | 5.20 (4.62, 5.83) |
| Others | 1,722 (28.76) | 28.74 (27.53, 29.99) | 1,744 (28.98) | 29.16 (27.94, 30.41) |
| **Type of healthcare received** | | | | |
| Inpatient care | 521 (8.70) | 9.40 (8.60, 10.20) | 518 (8.61) | 9.40 (8.60, 10.20) |
| Outpatient care | 5,466 (91.30) | 90.60 (89.80, 91.40) | 5,500 (91.39) | 90.60 (89.80, 91.40) |
| **Types of health facilities** | | | | |
| Public facilities | 1,024 (17.10) | 19.70 (18.70, 20.80) | 999 (16.60) | 19.2 (18.20, 20.30) |
| Private facilities | 3,937 (65.76) | 75.90 (74.70, 77.00) | 3,970 (65.97) | 76.4 (75.20, 77.50) |
| Others | 1,026 (17.14) | 4.40 (3.80, 4.90) | 1,049 (17.43) | 4.4 (3.90, 5.00) |
| **Waiting times for treatment** | | | | |
| <30 minutes | 5017 (83.80) | 81.30 (80.21, 82.34) | 5026 (83.52) | 80.93 (79.84, 81.97) |
| >30 minutes | 970 (16.20) | 18.70 (17.66, 19.79) | 992 (16.48) | 19.07 (18.03, 20.16) |
| **Consulted provider location** | | | | |

(*Continued*)

**Table 1.** (Continued)

| Participant characteristics | Male patients | | Female patients | |
|---|---|---|---|---|
| | Number of patients, n (%) | Utilisation of healthcare, % (95% CI) | Number of patients, n (%) | Utilisation of healthcare, % (95% CI) |
| *Rural based* | 2,719 (52.42) | 52.42 (51.06, 53.78) | 2,744 (52.79) | 52.79 (51.43, 54.15) |
| *Urban based* | 2,468 (47.58) | 47.58 (46.22, 48.94) | 2,454 (47.21) | 47.21 (45.85, 48.57) |
| **Overall** | 5,987 (49.87) | 86.64 (85.75, 87.48) | 6,018 (50.13) | 86.34 (85.45, 87.19) |

The majority of patients' highest rate of HCU was observed among those receiving outpatient care compared to that of inpatient care (90.60% vs. 9.40% for both genders). Health care utilisation was lower among those who had to wait more than 30 minutes for treatment than those who had to wait <30 minutes in both genders (81.30% vs. 18.7% for males; 80.93% vs. 18.03% for females). The HCU was highest among those who received care from private facilities than public facilities in both male (75.90% [95% CI: 74.70, 77.00] vs. 19.70% [95% CI: 18.70, 20.80]) and female patients (76.4% [95% CI: 75.20, 77.50] vs. 19.2% [95% CI: 18.20, 20.30]).

## Correlations of chronic disease-specific and gendered HCU

Table 2 presents the detail results of regression analysis using eight disease-specific different models (Model 1 to Model 8). In the final model (Model 8), patients who were diagnosed with chronic heart disease, diabetes, gastric/ulcer and chronic fever had a significant association with HCU for both genders (all p < 0.05).

We found that patients with chronic heart disease had significantly higher HCU compared to patients with other chronic diseases. However, the magnitude of association was higher among male patients (OR = 2.22; 95% CI:1.51–3.26; *p<0.001*) than their female counterparts

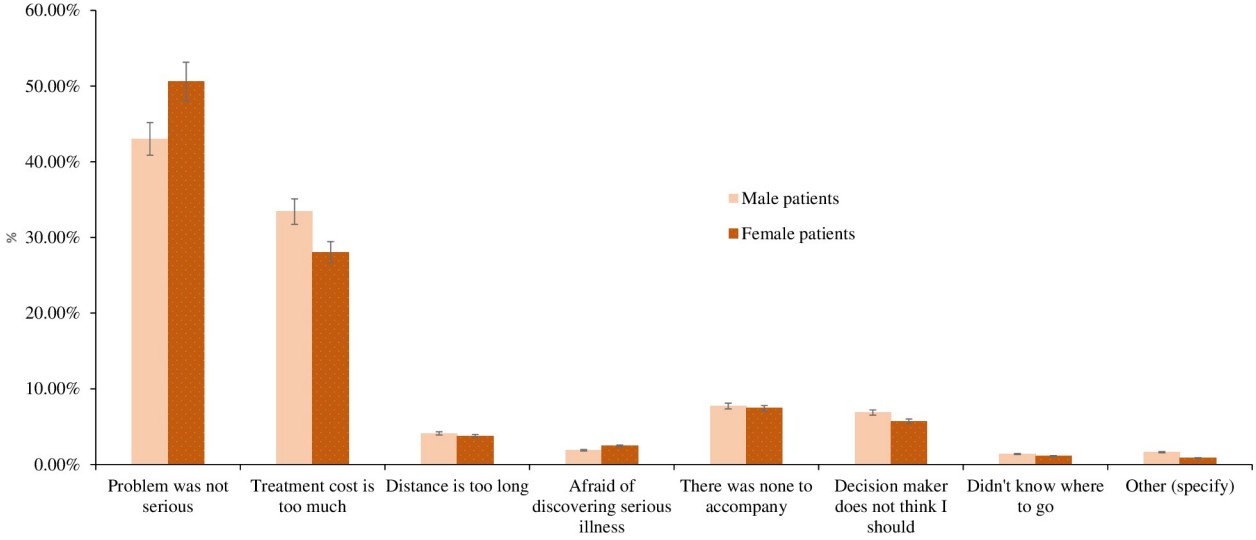

Figure 3. Percentage of not seeking any treatment

**Fig 3. Percentage of not seeking any treatment.**

**Table 2. Association between chronic disease-specific health care utilisation and related factors, by gender.**

| Chronic diseases or conditions | Model-1 | | Model-2 | | Model-3 | | Model-4 | |
|---|---|---|---|---|---|---|---|---|
| | Odds Ratio (95% CI) | p-value | Odds Ratio (95% CI) | p-value | Odds Ratio (95% CI) | p-value | Odds Ratio (95% CI) | p-value |
| **For male patients** | | | | | | | | |
| Chronic heart disease | 2.25 (1.53–3.31) | <0.001 | 2.22 (1.51–3.27) | <0.001 | 2.23 (1.52–3.27) | <0.001 | 2.23 (1.52–3.28) | <0.001 |
| Respiratory diseases/Asthma/Bronchitis | 1.36 (1.03–1.80) | 0.030 | 1.40 (1.06–1.84) | 0.019 | 1.40 (1.06–1.85) | 0.018 | 1.40 (1.06–1.85) | 0.018 |
| Gastric/ulcer | 0.75 (0.60–0.93) | 0.009 | 0.77 (0.62–0.96) | 0.018 | 0.77 (0.62–0.95) | 0.017 | 0.77 (0.62–0.95) | 0.017 |
| Blood pressure | 1.11 (0.82–1.51) | 0.490 | 1.14 (0.84–1.54) | 0.400 | 1.15 (0.85–1.55) | 0.381 | 1.15 (0.85–1.56) | 0.367 |
| Arthritis/Rheumatism | 1.00 (0.78–1.28) | 1.000 | 1.04 (0.81–1.33) | 0.749 | 1.04 (0.81–1.33) | 0.749 | 1.04 (0.82–1.33) | 0.746 |
| Diabetes | 1.79 (1.19–2.69) | 0.006 | 1.81 (1.20–2.74) | 0.004 | 1.82 (1.21–2.74) | 0.004 | 1.82 (1.21–2.74) | 0.004 |
| Chronic fever | 0.47 (0.36–0.62) | <0.001 | 0.49 (0.38–0.65) | <0.001 | 0.50 (0.38–0.65) | <0.001 | 0.50 (0.38–0.65) | <0.001 |
| Others (= reference group) | 1.00 | | 1.00 | | 1.00 | | 1.00 | |
| **For female patients** | | | | | | | | |
| Chronic heart disease | 1.45 (1.03–2.06) | 0.034 | 1.43 (1.01–2.03) | 0.042 | 1.43 (1.01–2.03) | 0.043 | 1.43 (1.01–2.03) | 0.042 |
| Respiratory diseases/Asthma/Bronchitis | 1.16 (0.89–1.51) | 0.267 | 1.19 (0.91–1.55) | 0.196 | 1.19 (0.91–1.55) | 0.197 | 1.19 (0.91–1.55) | 0.194 |
| Gastric/ulcer | 0.74 (0.60–0.92) | 0.007 | 0.77 (0.62–0.95) | 0.017 | 0.77 (0.62–0.96) | 0.019 | 0.77 (0.62–0.96) | 0.019 |
| Blood pressure | 0.93 (0.70–1.23) | 0.587 | 0.96 (0.72–1.27) | 0.771 | 0.97 (0.73–1.28) | 0.807 | 0.96 (0.73–1.28) | 0.801 |
| Arthritis/Rheumatism | 1.05 (0.82–1.34) | 0.702 | 1.10 (0.86–1.41) | 0.458 | 1.10 (0.86–1.41) | 0.450 | 1.10 (0.86–1.41) | 0.459 |
| Diabetes | 1.92 (1.26–2.94) | 0.002 | 1.94 (1.27–2.96) | 0.002 | 1.95 (1.28–2.98) | 0.002 | 1.95 (1.27–2.97) | 0.002 |
| Chronic fever | 0.45 (0.34–0.59) | <0.001 | 0.45 (0.34–0.60) | <0.001 | 0.46 (0.35–0.60) | <0.001 | 0.46 (0.35–0.61) | <0.001 |
| Others (= reference group) | 1.00 | | 1.00 | | 1.00 | | 1.00 | |

| Chronic diseases or conditions | Model-5 | | Model-6 | | Model-7 | | Model-8 | |
|---|---|---|---|---|---|---|---|---|
| | Odds Ratio (95% CI) | p-value | Odds Ratio (95% CI) | p-value | Odds Ratio (95% CI) | p-value | Odds Ratio (95% CI) | p-value |
| **For male patients** | | | | | | | | |
| Chronic heart disease | 2.22 (1.51–3.27) | <0.001 | 2.22 (1.51–3.27) | <0.001 | 2.22 (1.51–3.26) | <0.001 | 2.22 (1.50–3.26) | <0.001 |
| Respiratory diseases/Asthma/Bronchitis | 1.40 (1.06–1.85) | 0.018 | 1.40 (1.06–1.85) | 0.018 | 1.40 (1.06–1.85) | 0.018 | 1.40 (1.06–1.85) | 0.018 |
| Gastric/ulcer | 0.77 (0.62–0.96) | 0.017 | 0.77 (0.62–0.95) | 0.017 | 0.77 (0.62–0.98) | 0.017 | 0.77 (0.62–0.95) | 0.017 |
| Blood pressure | 1.15 (0.85–1.56) | 0.368 | 1.15 (0.85–1.56) | 0.366 | 1.15 (0.85–1.56) | 0.373 | 1.15 (0.85–1.56) | 0.372 |
| Arthritis/Rheumatism | 1.04 (0.81–1.33) | 0.753 | 1.04 (0.81–1.33) | 0.752 | 1.04 (0.81–1.32) | 0.775 | 1.04 (0.81–1.32) | 0.777 |
| Diabetes | 1.82 (1.21–2.74) | 0.004 | 1.82 (1.21–2.74) | 0.004 | 1.82 (1.21–2.75) | 0.004 | 1.82 (1.21–2.74) | 0.004 |
| Chronic fever | 0.49 (0.38–0.65) | <0.001 | 0.49 (0.38–0.65) | <0.001 | 0.49 (0.38–0.65) | <0.001 | 0.49 (0.38–0.65) | <0.001 |
| Others (= reference group) | 1.00 | | 1.00 | | 1.00 | | 1.00 | |
| **For female patients** | | | | | | | | |
| Chronic heart disease | 1.43 (1.01–2.02) | 0.045 | 1.43 (1.01–2.03) | 0.042 | 1.44 (1.02–2.04) | 0.040 | 1.44 (1.02–2.04) | 0.041 |
| Respiratory diseases/Asthma/Bronchitis | 1.19 (0.91–1.55) | 0.198 | 1.19 (0.92–1.56) | 0.188 | 1.20 (0.92–1.56) | 0.183 | 1.20 (0.92–1.56) | 0.179 |
| Gastric/ulcer | 0.77 (0.62–0.96) | 0.021 | 0.78 (0.63–0.97) | 0.024 | 0.78 (0.63–0.97) | 0.025 | 0.78 (0.63–0.97) | 0.026 |
| Blood pressure | 0.96 (0.73–1.28) | 0.795 | 0.97 (0.73–1.28) | 0.819 | 0.97 (0.73–1.29) | 0.839 | 0.97 (0.74–1.29) | 0.860 |
| Arthritis/Rheumatism | 1.09 (0.85–1.40) | 0.486 | 1.10 (0.86–1.41) | 0.458 | 1.10 (0.86–1.41) | 0.451 | 1.10 (0.86–1.41) | 0.444 |
| Diabetes | 1.94 (1.27–2.97) | 0.002 | 1.96 (1.28–3.00) | 0.002 | 1.96 (1.28–3.00) | 0.002 | 1.97 (1.29–3.01) | 0.002 |
| Chronic fever | 0.46 (0.35–0.61) | <0.001 | 0.46 (0.35–0.61) | <0.001 | 0.46 (0.35–0.61) | <0.001 | 0.46 (0.35–0.62) | <0.001 |
| Others (= reference group) | 1.00 | | 1.00 | | 1.00 | | 1.00 | |

Note: Model 1: Chronic illness or conditions; Model 2: Adjusted for Model-1 + type of health care; Model 3: adjusted for Model 2+ age, Model 4: adjusted for Model 3 + educational background; Model 5: adjusted for Model 4 + religion status; Model 6: adjusted for Model 5 + marital status; Model 7: adjusted for Model 6 + employment status; Model 8: adjusted for Model 7 + type of health care facilities, consultation provider's location, number of chronic comorbid conditions and waiting times. In the unadjusted model (results not shown in Table 2), all selected variables were significant at 5% or less risk level.

**Table 3. Sensitivity analysis testing robustness of results using the bootstrapping approach by resampling observations.**

| Chronic diseases or conditions | Observed Odds Ratio | Bootstrap Std. Err. | Normal-based 95% confidence interval | | P-value |
|---|---|---|---|---|---|
| **For male patients** | | | | | |
| Chronic heart disease | 2.21 | 0.44 | 1.49 | 3.29 | <0.001 |
| Respiratory diseases/Asthma/Bronchitis | 1.40 | 0.20 | 1.05 | 1.86 | 0.020 |
| Gastric/ulcer | 0.76 | 0.08 | 0.61 | 0.95 | 0.018 |
| Blood pressure | 1.14 | 0.18 | 0.84 | 1.56 | 0.378 |
| Arthritis/Rheumatism | 1.03 | 0.13 | 0.8 | 1.32 | 0.778 |
| Diabetes | 1.82 | 0.39 | 1.19 | 2.78 | 0.006 |
| Chronic fever | 0.49 | 0.06 | 0.37 | 0.64 | <0.001 |
| Other chronic diseases (= reference group) | 1.00 | | | | |
| Number of observations | 5,987 patients | | | | |
| Replications | 10,000 times | | | | |
| Wald chi$^2$ (p-value) | 112.15 (p<0.001) | | | | |
| Chronic diseases or conditions | Observed Odds Ratio | Bootstrap Std. Err. | Normal based 95% confidence interval | | P-value |
| **For female patients** | | | | | |
| Chronic heart disease | 1.44 | 0.27 | 1.00 | 2.07 | 0.048 |
| Respiratory diseases/Asthma/Bronchitis | 1.20 | 0.16 | 0.92 | 1.56 | 0.185 |
| Gastric/ulcer | 0.78 | 0.09 | 0.62 | 0.97 | 0.028 |
| Blood pressure | 0.97 | 0.14 | 0.73 | 1.30 | 0.842 |
| Arthritis/Rheumatism | 1.10 | 0.14 | 0.86 | 1.41 | 0.452 |
| Diabetes | 1.96 | 0.43 | 1.27 | 3.02 | 0.002 |
| Chronic fever | 0.46 | 0.07 | 0.35 | 0.61 | <0.001 |
| Other chronic diseases (= reference group) | 1.00 | | | | |
| Number of observations | 6,018 patients | | | | |
| Replications | 10,000 times | | | | |
| Wald chi$^2$ (p-value) | 109.61 (p<0.001) | | | | |

(OR = 1.44; 1.02–2.04; *p = 0.041*). Similarly, diabetes patients reported significant HCU in both genders (OR = 1.82; 1.21–2.74; *p = 0.004* for male patients and OR = 1.97, 1.28–3.00; *p = 0.002* for female patients). The magnitude of HCU also depended on the severity of diseases. For example, patients with gastric/ulcer had significantly lower HCU [23% for male patients, (OR = 0.77; 0.62–0.95; *p = 0.017*) or 22% for female patients, (OR = 0.78; 0.63–0.97; *p = 0.026*)] compared to patients diagnosed with other diseases. A similar association was observed for patients diagnosed with chronic fever (for male patients, OR = 0.49; 0.38–0.65; p = 0.026 or female patients, OR = 0.46; 0.35–0.61; *p<0.001*). In addition, these associations were consistent with sensitivity analysis testing robustness of results using the bootstrapping approach by resampling observations with 10,000 replications (Table 3).

## Discussion

This study examined the disease-stratified and gender- differentiated HCU in Bangladesh among patients with chronic diseases. The major chronic diseases reported were chronic heart diseases, respiratory diseases/asthma/bronchitis, gastric/ulcer, blood pressure, arthritis/rheumatism, diabetes, and chronic fever. They were found alike among males and females. However, the magnitude of seeking healthcare services due to these chronic conditions varied across the types of chronic diseases and gender. For example, participants with chronic heart

disease, diabetes, and respiratory diseases reported highest HCU in both genders, while chronic fever had the lowest HCU.

The seeking of healthcare services may be influenced by disease severity and various demographic and socioeconomic factors. The burden of chronic diseases combined with frequent acute illness episodes increases the risk of high levels of long-term adverse events (e.g., comorbidity, mortality and disability) compared to other diseases [5, 8, 11, 29]. Patients with chronic illness also have a greater risk of being diagnosed with other associated comorbidities, which increase utilisation of healthcare services [30]. Chronic diseases damage lives and adversely affect the quality of life and ultimate disability, which increases the HCU among affected patients [31]. Long-lasting chronic conditions result in a continuation of treatment and care, which increases the use of healthcare resources (e.g., specialist consultations, diagnostic, medicines) [32, 33]. The severity of the chronic illness (i.e. heart disease, diabetes) leads to more health care service utilisation, increasing the economic burden compared to the other diseases [16].

Taking a gendered perspective to disease-specific chronic illnesses shows the magnitude of HCU differs significantly between males and females. For instance, male patients with chronic heart disease utilised healthcare services at a rate more than two times higher compared to female patients. This trend was also consistent for respiratory diseases and blood pressure, although females slightly utilised more healthcare services for diabetes. Generally, the lower utilisation of healthcare among females in Bangladesh compared to males mainly depends on who is making the HCU decision, financial capability and accessibility power, knowledge and awareness [34–40]. One study reported that Bangladeshi males are more unwilling to adhere to and continue treatment for chronic heart disease compared to females [41]. These practices among males may lead to the recurrence of chronic heart diseases like hypertension and trigger more HCU for heart diseases. Another study in Bangladesh showed that the male-headed family heritage leads to demotivated women's decisions about their healthcare even when there is agreement from senior family members, especially husbands and/or mothers-in-law [42]. This supports the finding of another study [43] that showed that 37% of Bangladesh women had no decision-making power about their healthcare utilisation. This was even more extreme at 55.6% as reported in India [44]. However, studies in Spain and the USA reported that older female were more likely to use medical practitioners, outpatient health services and medications than men [45, 46]. In the USA, women face a higher rate of disability and poor health conditions and are more unlikely to receive the prescribed drugs due to cost [46, 47]. Besides, males have to pay much more for HCU due to the higher rate of obesity and cardiovascular diseases [48]. In addition, the location of healthcare service providers, disproportionate population density, and education are crucial factors for seeking healthcare [49–51]. Access to money for their own healthcare is an influential factor for women's HCU. A study expressed that only 14% of married women can decide on their health care in Ethiopia, while only 38% can use money independently for their healthcare [35]. Studies in India [52, 53] reported high (about 50%) gender disparity in healthcare expenditure which increased for older patients and also women's healthcare needs are regularly and often neglected or have less priority in households.

In the UK, women who get support from their husbands in decision making had a higher odds of HCU and it increased in urban areas compared to rural areas [27]. Another study in the UK showed that males are 32% less likely to have a primary healthcare consultation than females [54]. The cultural consequences exposed that the HCU may depend on disease severity and magnitude of health burden, especially for women. It is not necessarily true that the males are more conscious and knowledge enriched about the chronic diseases in Bangladesh that may influence the overall lower HCU.

Nevertheless, during recent years, remarkable success and changes have occurred in maternal HCU in Bangladesh. However, there are still low HCU and women's autonomy practices, particularly in rural areas, and for women with lower education and socioeconomic status [40, 55, 56]. The above evidence indicates that being a South Asian country, there exists a significant gap in opportunities and privileges for women in Bangladeshi families. Social supports, risk-pooling mechanisms, early risk detections and community-based awareness programs may contribute to achieving universal health coverage for women over time.

The current study utilised the most recent household income and expenditure survey data which is nationally representative of the Bangladeshi population. These national-level data make the study findings more precise and reliable. However, there are still some limitations regarding this study which the authors acknowledge. For instance, self-reported data of the key variables of interest were used, and findings should be interpreted cautiously. In addition, the survey data consist of information about self-reported illness, utilisation of healthcare services, and expenditure that might be affected by recall bias, although only information from the last 30 days was considered, which reduces the chances of potential recall bias due to the short recall period. Cross-sectional studies are normally a type of observational study design rather than longitudinal design; therefore, it is difficult to determine any causal relationships among variables. Moreover, in the HIES data, a high number of people reported "other" chronic disease without specifying the type of disease they suffered from. Therefore, we had to consider the most prevalent chronic diseases they suffered from.

## Conclusions

The present study focused on chronic disease-stratified and gender-based HCU in Bangladesh. HCU due to chronic illness is significantly higher among the male population than females. The circumstances demand that affordable and accessible healthcare services are urgently needed for women, especially in rural areas. The government and other related organisations should focus on improved healthcare system planning, healthcare service quality improvement strategies and special healthcare benefits for disadvantaged individuals, especially women. Social supports, risk-pooling mechanisms, early risk detection and community-based awareness development may contribute to progressing universal health coverage. In addition, resource allocation, capacity building, technology enabled health system can be considered to cope with the new challenges during this current pandemic and post-COVID healthcare management. Further rigorous research should be conducted to understand the core factors, exchange and enhance the beliefs and knowledge about chronic diseases and their gendered treatment in Bangladesh.

## Acknowledgments

This research was carried out using the 2016–2017 Bangladesh Household Income and Expenditure Survey. We would like to thank the HIES program for providing access to the data utilised in this research. We would also like to gratefully acknowledge the study's participants, reviewers and the academic editors of our manuscript.

## Author Contributions

**Conceptualization:** Rashidul Alam Mahumud, Sabuj Kanti Mistry.

**Data curation:** Rashidul Alam Mahumud.

**Formal analysis:** Rashidul Alam Mahumud.

**Investigation:** Rashidul Alam Mahumud, Md Parvez Mosharaf, Satyajit Kundu, Md. Ashfikur Rahman, Sabuj Kanti Mistry.

**Methodology:** Rashidul Alam Mahumud, Md Parvez Mosharaf.

**Project administration:** Rashidul Alam Mahumud.

**Resources:** Rashidul Alam Mahumud, Jeff Gow.

**Software:** Rashidul Alam Mahumud.

**Supervision:** Jeff Gow, Khorshed Alam.

**Validation:** Rashidul Alam Mahumud, Jeff Gow, Md. Ashfikur Rahman, Natisha Dukhi, Md Shahajalal, Sabuj Kanti Mistry, Khorshed Alam.

**Visualization:** Rashidul Alam Mahumud, Jeff Gow, Md. Ashfikur Rahman, Natisha Dukhi, Sabuj Kanti Mistry.

**Writing – original draft:** Rashidul Alam Mahumud, Md Parvez Mosharaf, Satyajit Kundu, Md. Ashfikur Rahman, Md Shahajalal.

**Writing – review & editing:** Rashidul Alam Mahumud, Jeff Gow, Md Parvez Mosharaf, Satyajit Kundu, Md. Ashfikur Rahman, Natisha Dukhi, Md Shahajalal, Sabuj Kanti Mistry, Khorshed Alam.

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
