## [Decision Letter · Decision Letter 0]

15 Jul 2022

PONE-D-22-15927The emerging burden of chronic diseases, disease-stratified exploration and gender-differentiated healthcare utilisation among adult patients in BangladeshPLOS ONE

Dear Dr. Mahumud,

Thank you for submitting your manuscript to PLOS ONE. After careful consideration, we feel that it has merit but does not fully meet PLOS ONE’s publication criteria as it currently stands. Therefore, we invite you to submit a revised version of the manuscript that addresses the points raised during the review process.

Thanks for  conducting this  study  ; Please consider some  important points  in revised version

1- House Hold Survey should be  more clear  in method section

2- Please elaborate  employed analysis  for readers

We look forward to receiving your revised manuscript.

Kind regards,

Hamid Reza Baradaran, M.D., Ph.D.,

Academic Editor

PLOS ONE

Journal Requirements:

Additional Editor Comments:

Thanks for conducting this study ; Please consider some important points in revised version

1- House Hold Survey should be more clear in method section

2- Please elaborate employed analysis for readers

Reviewers' comments:

Reviewer's Responses to Questions

**Comments to the Author**

1. Is the manuscript technically sound, and do the data support the conclusions?

Reviewer #1: Partly

Reviewer #2: Partly

2. Has the statistical analysis been performed appropriately and rigorously? 

Reviewer #1: Yes

Reviewer #2: No

3. Have the authors made all data underlying the findings in their manuscript fully available?

Reviewer #1: Yes

Reviewer #2: No

4. Is the manuscript presented in an intelligible fashion and written in standard English?

Reviewer #1: Yes

Reviewer #2: No

5. Review Comments to the Author

Reviewer #1: Title: OK however a shorter title is desirable.

abstract: OK; however, a number of revisions may be necessary in case of revision of the following sections as requested afterward.

introduction: OK

material & method: Please notice the the following comments:

- Please address the matter of ethical issues and their confirmation in this section as well.

- please define the inclusion and exclusion criteria in more detail. for instance, it is necessary to clarify if more than 1 person were permitted o be recruited for study in each household. this is very important for estimating the variance of the proposed measures.

- I think it is necessary to show the reliability of the responses to the utilization question too.

- Please state how disease history variables were measured in each individuals explicitly. this will affect the justification of the next section (results).

- Considering the type of sampling please define how variables were weighted for the analysis. because of considering all members in each household for the study, there is a considerable amount of intra-class correlation which will affect the precision of the estimations.

- I think this level of significance is very conservative to recruit variables in the next stage model. (line 8, page 7)

- In the table 1, I think it is better to show this table according to utilization instead of gender. in this case it will be more sensible to see the association of the each variable and utilization separately.

Reviewer #2: Introduction:

- In the introduction, there is no evidence that the burden of chronic diseases in Bangladesh is increasing in the last decade. These diseases are almost age-related. The introduction does not mention the prevalence of chronic diseases at different ages or age-adjusted rates.

- In the introduction, it should be noted whether the population of Bangladesh is aging, which has caused concern?

- The importance of conducting this research is not clear.

Methods:

- Data and information in the study is related to 2016-2017. My question is how to deal with the emerging burden of disease? In my opinion, if the data were used from a cohort study or from two consecutive censuses, the results would be more accurate and comprehensive.

- Where can I find a list of chronic diseases? For example, in this section, there is no mentioned to kidney diseases.

- The method of adjusting variables in models is confusing and unclear.

- Was regression or logistic regression used in this study?

- The method of initial selection of variables to enter the model is not clear.

- The level of statistical significance should be less than 5% and not equal to 5%.

Results:

- Percentages should be written in numbers and not in letters.

- The title of the article states that the study was conducted on adults, but the results indicate that almost 42% of people are young. This is a contradiction.

- If one-third of patients had no formal education, how do they name their disease? The respondent is not known in this study.

- In this section was stated that “The utilisation of health care reduced as patients aged”. Has a statistical test been performed?

- In the division of chronic diseases (see Table 1), there is a group called "others", which is about 28% of patients. I think this group should be divided into more specific subgroups.

Final:

- This manuscript does not have enough coherence. The title of the study is not in line with the final results and conclusions. Therefore, it cannot be published.

6. PLOS authors have the option to publish the peer review history of their article (what does this mean?). If published, this will include your full peer review and any attached files.

Reviewer #1: **Yes: **Babak Eshrati

Reviewer #2: No

---

## [Author Response · Author response to Decision Letter 0]

20 Sep 2022

Date: 21 Aug, 2022 

Prof. Hamid Reza Baradaran

Academic Editor 

PLOS ONE

Response to Reviewers’ comments on Manuscript Number: PONE-D-22-15927

The revised title of the manuscript: "The burden of chronic diseases, disease-stratified exploration and gender-differentiated healthcare utilisation among patients in Bangladesh".

Dear Prof. Baradaran

Thank you for the opportunity to revise our manuscript. We found the reviewers' comments and feedback very helpful in improving the manuscript and we have revised the manuscript accordingly. Please find below our point–by–point responses to each of the reviewers' comments. Two versions of our paper – one clean copy, and one with marked-up copy showing the changes made – are submitted. 

Yours sincerely,

Dr Rashidul Alam Mahumud (corresponding author)

Response to the Editor Comments

Additional Editor Comments:

Thanks for conducting this study; Please consider some important points in revised version

1- Household survey should be more clear in method section

2- Please elaborate employed analysis for readers

Authors’ response: Thank you very much for your comments and suggestions regarding the manuscript. We have updated the sections in the revised manuscript.

Response to Reviewer #1

1. Title: OK however a shorter title is desirable.

Author’s response: We have revised the title of this manuscript. Currently it reads, “The burden of chronic diseases, disease-stratified exploration and gender-differentiated healthcare utilisation among patients in Bangladesh”. Please see the revised title. 

2. abstract: OK; however, a number of revisions may be necessary in case of revision of the following sections as requested afterward.

Author’s response: Thanks for your comment. The entire manuscript has been revised and updated to incorporate the reviewer’s comments. Please see the revised manuscript.

3. Introduction: OK

Author’s response: Thank you.

4. Material & method: Please notice the following comments: Please address the matter of ethical issues and their confirmation in this section as well.

Author’s response: Thank you for the suggestion. We have added sub-section about ethical issues in the method section. Please see the revised manuscript (page 9, lines 6-9).

5. Please define the inclusion and exclusion criteria in more detail. For instance, it is necessary to clarify if more than 1 person were permitted to be recruited for study in each household. this is very important for estimating the variance of the proposed measures.

Author’s response: This study used the nationally representative Household Income and Expenditure Survey, 2016-17 data. 

The participants for the present analysis were selected based on the HIES 2016-17 survey protocol and the following inclusion criteria: i) an individual who had suffered from any chronic disease for the last 12 months or more, ii) an individual who suffered from any chronic diseases or chronic conditions in the last 30 days, and iii) individual who received any treatment due to chronic illness in the last 30 days. This study used individual-level data to explore who has been exposed to any chronic disease. Please see page 6 (lines 19-24).

6. I think it is necessary to show the reliability of the responses to the utilisation question too.

Authors’ response: We analysed secondary data from the nationally representative Household Income and Expenditure Survey (HIES 2016-17). Bangladesh Bureau of Statistics (BBS) has already validated the study settings and tested the reliability of the data. The details of the study settings, questionnaire, and quality control measures have been described in the HIES 2016-17 report summary1. The HIES 2016-17 survey was based on an established protocol1. The HIES is a cross-sectional survey conducted by the BBS in Bangladesh every five (5) years since 19731; throughout the period of implementation, the HIES tools have been thoroughly reviewed to address the validity and reliability of the results. 

In line with the objective of HIES survey, the HIES 2016-17 survey collected information under nine modules: 1) household information, 2) education, 3) health: illnesses and injuries, 4) economic activities and wage employment, 5) non-agricultural enterprises, 6) housing, 7) agriculture, 8) other assets and income, and 9) consumption. However, the objective of the current article was to investigate disease-stratified healthcare utilisation (HCU) among Bangladesh patients with chronic diseases from a gender perspective. Therefore, we only use the indicators pertaining to chronic disease and health service utilisation along with socio-demographic characterises of the participants.

The HIES datasets are widely accepted and validated to produce scientific evidence. It is also used for monitoring the progress of poverty reduction and the Sustainable Development Goals (SDGs) indicators in Bangladesh. Please see some recent existing publications using the HIES data sets.

1. Ahmed et al. (2022). Assessing the incidence of catastrophic health expenditure and impoverishment from out-of-pocket payments and their determinants in Bangladesh: evidence from the nationwide Household Income and Expenditure Survey 2016. International Health; 14: 84–96. https://doi.org/10.1093/inthealth/ihab015

2. Rahman et al. (2022). Financial risk protection in health care in Bangladesh in the era of Universal Health Coverage. PLoS ONE 17(6): e0269113. https://doi.org/10.1371/journal.pone.0269113

3. Hossain et al. (2022). Do the issues of religious minority and coastal climate crisis increase the burden of chronic illness in Bangladesh? BMC Public Health 22(270) https://doi.org/10.1186/s12889-022-12656-5

4. Mitu et al. (2022). Spatial Differences in Diet Quality and Economic Vulnerability to Food Insecurity in Bangladesh: Results from the 2016 Household Income and Expenditure Survey. Sustainability 2022, 14(5643). https://doi.org/10.3390/su14095643

5. Mishra et al. (2015). Abiotic stress and its impact on production efficiency: The case of rice farming in Bangladesh. Agriculture, Ecosystems and Environment 199; 146–153. http://dx.doi.org/10.1016/j.agee.2014.09.006

This study does not require any additional reliability test for healthcare utilisation questions. Please see more detail information in the published HIES 2016-17 report. Please see page 6 (lines 1-22).

7. Please state how disease history variables were measured in each individual explicitly. this will affect the justification of the next section (results).

Author’s response: Thank you for your insightful feedback. 

Our study population was selected based on inclusion criteria: i) an individual who had suffered from any chronic disease for the last 12 months or more, ii) individual who received any treatment due to chronic illness in the last 30 days, and iii) individual who received any treatment due to chronic illness in the last 30 days.

Participants responded based on their disease diagnosis, experiences, symptoms of illness and course of treatment. All health-related information was self-reported in the HIES survey. For conducting the analysis, we used chronic disease-related questions that were included in Module-3: Health (Illnesses and Injuries) of the primary survey. For example, while collecting chronic disease-related information, the enumerators were instructed to ask the respondents, "Have you suffered from any chronic illness/disability in the last 12 months or more?" (if yes); then, participants were asked a second question “What chronic illness/disability are you suffering from?” with response options: 1) chronic fever, 2) Injuries/disability, 3) Chronic heart disease, 4) Respiratory Diseases/ Asthma/Bronchitis, 5) Diarrhoea/dysentery, 6) Gastric/ulcer, 7) Blood pressure, 8) Arthritis/Rheumatism, 9) Skin problem, 10) Diabetes, 11) Cancer, 12) Kidney diseases, 13) Liver Diseases, 14) Mental Health, 15) Paralysis, 16) Ear/ENT problem, 17) Eye problem, or 18 other (specify). 

8. Considering the type of sampling please define how variables were weighted for the analysis. because of considering all members in each household for the study, there is a considerable amount of intra-class correlation which will affect the precision of the estimations.

Author’s response: Thanks for pointing this out. We agree with you. In our analysis, we investigated individual-level data to estimate disease-specific healthcare utilisation due to chronic illness by gender prospective. However, we have followed standard ‘svy’ prefix command to address sampling weights. In addition, before running the final model, we checked for multicollinearity using Variance inflation factor (VIF) among the selected variables, no serious issues multicollinearity were found among the variables (all variables with VIF <5.00). The model was tested for sensitivity using the bootstrapping approach by resampling observations with 10,000 replications. Please page 8-10 (new Table 3). 

9. I think this level of significance is very conservative to recruit variables in the next stage model. (Line 8, page 7)

Author’s response: We followed the standard procedure of independent variable selection for the model with the widely used cut-off of p ≤ 0.05 in the unadjusted model [1–5]. 

10. In the table 1, I think it is better to show this table according to utilisation instead of gender. in this case it will be more sensible to see the association of each variable and utilisation separately.

Author’s response: Thank you. Our study objective was to investigate disease-stratified healthcare utilisation (HCU) among patients with chronic diseases in Bangladesh from a gender perspective. Therefore, Table 1 has been formatted to report healthcare utilisations by gender. All analyses were stratified by gender according to the study objective. Please see Table 1.

Response to the Reviewer #2

1. Introduction: In the introduction, there is no evidence that the burden of chronic diseases in Bangladesh is increasing in the last decade. These diseases are almost age-related. The introduction does not mention the prevalence of chronic diseases at different ages or age-adjusted rates.

Author’s response: Thank you very much for your comment. We have updated the introduction part in the revised manuscript. Please check the revised manuscript. Please see page 4 and 5. 

2. In the introduction, it should be noted whether the population of Bangladesh is aging, which has caused concern?

Author’s response: Basically, the analytical dataset of this study reveals that chronic diseases are not only significantly prevalent among the aged population but also among the young and young adults (Please see Table 1). Therefore, in general, the burden of chronic disease is not only a major concern for the aged but also the young and young adult population in Bangladesh. Please see page 4 and 5. 

3. The importance of conducting this research is not clear.

Author’s response: The introduction has now been revised and the significance of the research has been incorporated in the manuscript. Please see the revised introduction section

This section now reads: “There are still gender disparities in decision-making, roles and rights at home, and self-esteem when it comes to empowering women, which limits their access to healthcare in developing countries [20-22]. This is also true for Bangladesh, where men are often viewed as the head of households, decision-makers and are usually in charge of household resources and who typically decide on the women’s health needs and where and when they should utilise healthcare services [20, 22]. To achieve SDG 5: Health and gender equality, it is imperative to ensure women have access to appropriate health care utilisation [20, 22]. However, there is scant information on existing gender disparities in utilising healthcare services among patients with chronic diseases in Bangladesh. As a result, this study aimed to examine the gender perspective of HCU among patients with chronic diseases in Bangladesh”. Please see page 5 (lines 7-24).

4. Methods: Data and information in the study is related to 2016-2017. My question is how to deal with the emerging burden of disease? In my opinion, if the data were used from a cohort study or from two consecutive censuses, the results would be more accurate and comprehensive.

Author’s response: The title has been revised to delete the terms “emerging”. It now reads, “The burden of chronic diseases, disease-stratified exploration and gender-differentiated healthcare utilisation among patients in Bangladesh”. 

5. Where can I find a list of chronic diseases? For example, in this section, there is no mentioned to kidney diseases.

Author’s response: Thank you for the insightful feedback. The main objective of this study was to examine disease-stratified healthcare utilisation (HCU) among Bangladesh patients with chronic diseases from a gender perspective. In the analytical exploration, the authors performed the disease-specific analysis of HCU by gender. For analysis, we recorded type of diseases based on the most reported but kidney diseases had a low frequency ones.

In this study, we did not report the prevalence for all diseases separately due to low frequency, including kidney diseases (Figure 2). Please see Figure 2.

6. The method of adjusting variables in models is confusing and unclear.

Author’s response: Before adjusting the variables in the final model 8, firstly we did the univariate analysis and then step-by-step multiple regression. That has been clearly described in the “Statistical Analysis” section. We have revised the methods section. Please see page 8-10. 

7. Was regression or logistic regression used in this study?

Author’s response: We performed Logistic Regression. Please see page 8.

8. The method of initial selection of variables to enter the model is not clear.

Author’s Response: For the analytical exploration, the choice of estimation approach was informed by the nature of the outcome variables under consideration in each model. To build the regression model, explanatory variables were selected based on published literature [17,25–29], available information of this dataset and explored bivariate relationships (unadjusted analysis) between variables. In our analysis, we investigated individual-level data to estimate disease-specific healthcare utilisation due to chronic illness by gender prospective. We followed the standard procedure of independent variables selection for a model with the widely used cut-off value (p ≤ 0.05) in the unadjusted model. That has been clearly described in the “Statistical Analysis” section. Please see page 8-10. 

9. The level of statistical significance should be less than 5% and not equal to 5%.

Author’s response: Thank you. Statistical significance was considered at ≤ 5% risk level. Please see page 10 (lines 7-8).

10. Results: Percentages should be written in numbers and not in letters.

Author’s response: It has now been written in numbers in the revised manuscript. Please see the revised manuscript. 

11. The title of the article states that the study was conducted on adults, but the results indicate that almost 42% of people are young. This is a contradiction.

Author’s response: We have revised the title of this manuscript to remove the word “adult”. Currently it reads, “The burden of chronic diseases, disease-stratified exploration and gender-differentiated healthcare utilisation among patients in Bangladesh”. Please see the revised title.

12. If one-third of patients had no formal education, how do they name their disease? The respondent is not known in this study.

Author’s response: Thank you for raising this issue. The present study used the HIES 2016-17 data. The survey procedures are comprehensively explained in the HIES 2016-17 final report. The survey was conducted by the Bangladesh Bureau of Statistics (BBS), which is one of the core national surveys implemented by the Bangladesh Government. The survey team formulated an operational definition of each of the variable used in the questionnaire and before conducting this survey, detailed training was provided to the enumerators and the supervisors on every aspect of the questionnaire including ways of collecting information, particularly from those having less educational attainments. The HIES survey tool was translated from English into the local (Bengali) language to ensure that participants understand the questions. 

For example, while collecting chronic disease-related information, the enumerators were instructed to ask the respondents, "Have you suffered from any chronic illness/disability in the last 12 months or more?" (if yes); then, participants were asked a second question “What chronic illness/disability are you suffering from?” with response options: 1) chronic fever, 2) Injuries/disability, 3) Chronic heart disease, 4) Respiratory Diseases/ Asthma/Bronchitis, 5) Diarrhoea/dysentery, 6) Gastric/ulcer, 7) Blood pressure, 8) Arthritis/Rheumatism, 9) Skin problem, 10) Diabetes, 11) Cancer, 12) Kidney diseases, 13) Liver Diseases, 14) Mental Health, 15) Paralysis, 16) Ear/ENT problem, 17) Eye problem, or 18 other (specify). Participants responded based on their disease diagnosis, experiences, symptoms of illness and course of treatment. In order to get valid information, the enumerators also probed where necessary, asked the respondents to show any relevant documents e.g., prescriptions and test reports, or explained to the respondents about chronic diseases using various case scenarios as outlined in the survey guidelines and instructions. Please see page 7-8.

13. In this section was stated that “The utilisation of health care reduced as patients aged”. Has a statistical test been performed?

Authors’ response: Thank you for your comment. The distribution of HCU for chronic diseases within the background characteristics of the participants by gender is presented in Table 1. We found that the utilisation of health care was reduced as patients aged, which only the descriptive data. We did not perform any statistical test, which is outside the scope of this research objective. 

14. In the division of chronic diseases (see Table 1), there is a group called "others", which is about 28% of patients. I think this group should be divided into more specific subgroups.

Authors’ response: Thank you. We have developed a new figure (Figure 2) to show the distribution of chronic diseases. In the analytical exploration, the authors’ performed disease-specific analysis in terms of HCU by gender. For conducting analysis, we recorded type of diseases based on the most privilege of reported diseases. However, in the HIES data, a high number of people reported “others” chronic disease without specifying the type of disease they suffered from. In this study, we did not report the prevalence of all diseases due to low frequency (Figure 2). Therefore, we have considered the most prevalent chronic diseases in our study.. We have added this information in the limitation. Please see the revise manuscript (page 15, lines 6-8).

15. Final: This manuscript does not have enough coherence. The title of the study is not in line with the final results and conclusions. Therefore, it cannot be published.

Authors’ response: We highly appreciate the reviewers' insightful and helpful comments on our manuscript. This manuscript has been revised substantially and we strongly believe that the comments and suggestions have increased the scientific value of revised manuscript in line standards. We believe you would reconsider your decision and accept our revision. Please see the revised manuscript.

References:

1. Hossain A, Alam MJ, Mydam J, Tareque M. Do the issues of religious minority and coastal climate crisis increase the burden of chronic illness in Bangladesh? BMC Public Health. 2022;22: 1–19. doi:10.1186/s12889-022-12656-5

2. Imtiaz A, Khan NM, Hasan E, Johnson S, Nessa HT. Patients’ choice of healthcare providers and predictors of modern healthcare utilisation in Bangladesh: Household Income and Expenditure Survey (HIES) 2016-2017 (BBS). BMJ Open. 2021;11: 1–10. doi:10.1136/bmjopen-2021-051434

3. Kastor A, Mohanty SK. Disease-specific out-of-pocket and catastrophic health expenditure on hospitalization in India: Do Indian households face distress health financing? PLoS One. 2018;13: 1–18. doi:10.1371/journal.pone.0196106

4. Sheikh N, Sarker AR, Sultana M, Mahumud RA, Ahmed S. Disease ‑ specific distress healthcare financing and catastrophic out ‑ of ‑ pocket expenditure for hospitalization in Bangladesh. 2022; 1–16. doi:10.1186/s12939-022-01712-6

5. Sultana M, Mahumud RA, Sarker AR. Burden of chronic illness and associated disabilities in Bangladesh: Evidence from the Household Income and Expenditure Survey. Chronic Dis Transl Med. 2017;3: 112–122. doi:10.1016/j.cdtm.2017.02.001

6. Ahmad S, Maqbool PA. Health Seeking Behaviour and Health Service Utilization in Lucknow. SSRN Electron J. 2013. doi:10.2139/ssrn.2326415

---

## [Decision Letter · Decision Letter 1]

5 Feb 2023

PONE-D-22-15927R1The burden of chronic diseases, disease-stratified exploration and gender-differentiated healthcare utilisation among patients in BangladeshPLOS ONE

Dear Dr. Mahumud,

Thank you for submitting your manuscript to PLOS ONE. After careful consideration, we feel that it has merit but does not fully meet PLOS ONE’s publication criteria as it currently stands. Therefore, we invite you to submit a revised version of the manuscript that addresses the points raised during the review process.

ACADEMIC EDITOR: Make sure to address the comments made by reviewer #1 in the relation to accessing original data as you need to address the required revisions. 

We look forward to receiving your revised manuscript.

Kind regards,

Tarik A. Rashid, PhD

Academic Editor

PLOS ONE

Journal Requirements:

Reviewers' comments:

Reviewer's Responses to Questions

**Comments to the Author**

1. If the authors have adequately addressed your comments raised in a previous round of review and you feel that this manuscript is now acceptable for publication, you may indicate that here to bypass the “Comments to the Author” section, enter your conflict of interest statement in the “Confidential to Editor” section, and submit your "Accept" recommendation.

Reviewer #1: (No Response)

Reviewer #2: All comments have been addressed

2. Is the manuscript technically sound, and do the data support the conclusions?

Reviewer #1: No

Reviewer #2: Yes

3. Has the statistical analysis been performed appropriately and rigorously? 

Reviewer #1: No

Reviewer #2: Yes

4. Have the authors made all data underlying the findings in their manuscript fully available?

Reviewer #1: No

Reviewer #2: Yes

5. Is the manuscript presented in an intelligible fashion and written in standard English?

Reviewer #1: Yes

Reviewer #2: Yes

6. Review Comments to the Author

Reviewer #1: Thanks for the reply of the distinguished authors. According to the reply letter and the revised paper, it seems the author do not appropriate access to the original data; so that the revised paper does not address the required revisions. For instance, the matter of recruitment of the individuals from the households, it is not clear how many people were recruited from each household so that the matter of intraclass correlation is not clarified in the response. this is true for other requested revisions by the two reviewers.

Reviewer #2: I have matched all the authors' corrections with my own comments. All comments have been addressed by authors. In my opinion, given the corrections made, this manuscript is publishable.

7. PLOS authors have the option to publish the peer review history of their article (what does this mean?). If published, this will include your full peer review and any attached files.

Reviewer #1: **Yes: **babak eshrati

Reviewer #2: No

---

## [Author Response · Author response to Decision Letter 1]

19 Feb 2023

Date: 20/02/2023 

Dr Tarik A. Rashid

Academic Editor 

PLOS ONE

Response to Reviewers’ comments on Manuscript Number: PONE-D-22-15927R2

The revised title of the manuscript: "The burden of chronic diseases, disease-stratified exploration and gender-differentiated healthcare utilisation among patients in Bangladesh".

Dear Dr. Rashid

Thank you for the opportunity to revise our manuscript. We found the reviewers' comments and feedback very helpful in improving the manuscript and we have revised the manuscript accordingly. Please find below our point–by–point responses to each of the reviewers' comments. In addition, we have changed and updated the references list to provide appropriate and correct references. As a result, some of the references has been changed. Two versions of our paper – one clean copy, and one with marked-up (TC) copy showing the changes made – are submitted. 

Yours sincerely,

Dr Rashidul Alam Mahumud (corresponding author)

Response to the Editor Comments

Journal Requirements:

Authors’ response: 

Thanks. We have checked all references and corrected them where appropriate and necessary. Please see the updated and corrected references list. 

Response to Reviewer #1

Reviewer #1: Thanks for the reply of the distinguished authors. According to the reply letter and the revised paper, it seems the author do not appropriate access to the original data; so that the revised paper does not address the required revisions. For instance, the matter of recruitment of the individuals from the households, it is not clear how many people were recruited from each household so that the matter of intraclass correlation is not clarified in the response. this is true for other requested revisions by the two reviewers.

Author’s Response: Thank you very much for your comment. 

We provided one section about the Availability of data and materials.

It reads: “Bangladesh Household Income and Expenditure Survey (HIES) is conducted by the Bangladesh Bureau of Statistics (BBS) with technical and financial support from the World Bank. This research was carried out using the 2016-2017 Bangladesh Household Income and Expenditure Survey. However, the BBS imposed legal restrictions that prevent the sharing of data publicly. Data can be shared upon request to the corresponding author with the permission of the BBS (Director General, Bangladesh Bureau of Statistics, dg@bbs.gov.bd, +88-02-5500-7056, www.bbs.gov.bd)”. Please see page 19 (lines 12-19).

We are more than happy to reply again this same comment as we did before. Our study was designed and structured based on examining disease-stratified healthcare utilisation (HCU) among Bangladesh patients with chronic diseases from a gender perspective. The entire dataset for this study consisting of 12,005 patients with diagnosed chronic diseases, has been collected from a nationally representative Household Income and Expenditure Survey (HIES), 2016-17 data. This dataset is open, well-structured, very popular, and rigorously used for different publications worldwide. In our study, we have selected the individuals from the households based on the following conditions: i) an individual who had suffered from any chronic disease for the last 12 months or more, ii) an individual who suffered from any chronic diseases or chronic conditions in the last 30 days, and iii) individual who received any treatment due to chronic illness in the last 30 days. 

This study used individual-level data (not focusing on the households) to explore who has been exposed to any chronic disease. Therefore, It was not necessary to keep track of how many households were occupied by the overall sample population, which is out of the scope of this study. Please see page 6 (lines 19-24).

Response to Reviewer #2

Reviewer #2: I have matched all the authors' corrections with my own comments. All comments have been addressed by the authors. In my opinion, given the corrections made, this manuscript is publishable.

Author’s Response: Thanks. All of the reviewer’s comments help improve the quality of this manuscript.

---

## [Editor Report · Decision Letter 2]

27 Mar 2023

The burden of chronic diseases, disease-stratified exploration and gender-differentiated healthcare utilisation among patients in Bangladesh

PONE-D-22-15927R2

Dear Dr. Mahumud,

We’re pleased to inform you that your manuscript has been judged scientifically suitable for publication and will be formally accepted for publication once it meets all outstanding technical requirements.

Kind regards,

Tarik A. Rashid, PhD

Academic Editor

PLOS ONE
---

## [Editor Report · Acceptance letter]

19 Apr 2023

PONE-D-22-15927R2 

The burden of chronic diseases, disease-stratified exploration and gender-differentiated healthcare utilisation among patients in Bangladesh 

Dear Dr. Mahumud:

I'm pleased to inform you that your manuscript has been deemed suitable for publication in PLOS ONE. Congratulations! Your manuscript is now with our production department. 

Kind regards, 

on behalf of

Dr. Tarik A. Rashid 

Academic Editor

PLOS ONE